# A Case of Advanced Tubal Ectopic Pregnancy after Emergency Contraception

**DOI:** 10.3390/healthcare10081590

**Published:** 2022-08-22

**Authors:** Stefano Restaino, Matilde Degano, Diana Padovani, Anna Biasioli, Valentina Capodicasa, Giuseppe Vizzielli, Lorenza Driul

**Affiliations:** 1Department of Obstetrics and Gynecology, Azienda Ospedaliera Universitaria Friuli Centrale, ASUFC Udine, 33100 Udine, Italy; 2Medical Area Department, DAME, Università degli Studi di Udine, Via delle Scienze, 206, 33100 Udine, Italy

**Keywords:** ectopic pregnancy, emergency contraception, ulipristal acetate

## Abstract

Ectopic pregnancy is a relatively common condition and an important cause of morbidity in women of childbearing age. The most frequent implantation site is the fallopian tube. Most cases are diagnosed in an early gestational period. Patients come to the attention of clinicians for pelvic pain and vaginal blood loss, and consequent diagnosis is made through clinical presentation, laboratory tests, and ultrasound. Other rarer implantation sites such as the abdominal cavity give space for ectopic pregnancy to grow until later gestational ages, delaying diagnosis. This is a rare case of a healthy 41-year-old woman with an advanced ectopic pregnancy after emergency contraception with Ulipristal Acetate. The patient went to visit for amenorrhea after taking a contraceptive. Evaluation with ultrasound demonstrated a 10 + 4 weeks’ unruptured tubal pregnancy with fetal heart rate. The patient underwent laparoscopic salpingectomy without complication. This is the first case of such an advanced ectopic pregnancy in a woman who performed emergency contraception with Ulipristal Acetate.

## 1. Introduction

Ectopic pregnancy is defined as a pregnancy that occurs outside the uterine cavity [1]. The correct incidence of ectopic pregnancies is difficult to establish. According to the Centers for Disease Control and Prevention, ectopic pregnancies account for approximately 2% of all reported pregnancies, but United States surveillance data have not been updated since 1992 [2]. Due to the risk of rupture, ectopic pregnancy is a major cause of maternal morbidity and occasionally mortality [1].

The most common implantation site of ectopic pregnancy is the fallopian tube. Usually, the rupture of that site happens in the first few weeks of gestation. Other implantation sites, including the abdomen, allow for further gestational development due to increased free space and the ability of the implantation site to distend [1].

In many cases, there are no clinical signs and no symptoms. The most frequent signs and symptoms are bleeding, abdominal pain, and amenorrhea. The clinical suspicious of ectopic pregnancy should be confirmed by transvaginal ultrasound and quantitative β-HCG level. The ultrasound appearance of an ectopic pregnancy depends on the stage of development at which the examination is performed. In the case of a tubal pregnancy, we can find an empty ectopic gestational sac (tubal ring), an inhomogeneous adnexal mass (blob sign), a pelvic effusion or a pregnancy of uncertain localization. More rarely, it is possible to visualize an embryo in an ectopic site. In a patient with a suspected ectopic pregnancy, the finding of a non-cystic adnexal mass has a sensitivity of 84% and a specificity of 98.9% [3]. Measurement of serum β-HCG trend helps in the diagnosis in association with ultrasound.

In case of early diagnosis with no evidence of rupture and low values of β-HCG, medical therapy with methotrexate is feasible in many women. Methotrexate is a folate antagonist that interrupts the synthesis of purine nucleotides and the amino acids serine and methionine. Since it inhibits DNA synthesis and repair, this drug predominantly affects actively proliferating tissue such as trophoblastic tissue [4]. β-HCG concentration is the most important factor determining the failure of the medical treatment, as a low β-HCG value leads to a high success rate of Methotrexate [1]

A surgical approach is required when a patient with ectopic pregnancy presents hemodynamic instability, symptoms of an ongoing rupture of the ectopic mass, signs of intraperitoneal bleeding or in suspicion of a heterotopic pregnancy. Moreover, surgery is necessary when a patient meets absolute contraindications to medical management [1]. Surgical treatment is generally performed using laparoscopic salpingectomy or laparoscopic salpingotomy. Salpingectomy is the preferred approach, but salpingotomy should be chosen in patients with history of fertility-reducing factors (for example in the case of a damaged contralateral tuba) [1].

Patients should be informed that the incidence of recurrent ectopic pregnancy is approximately 18%, with no substantial variations between different treatments, unless there is a previous history of subfertility. In this case, the outcome is improved with conservative or medical treatments [1]. 

Ectopic pregnancy can also occur if emergency contraception fails. Emergency contraception is a method used to prevent pregnancy in the days following an unprotected sexual intercourse. It includes emergency contraception pills (ECPs) and copper-earing intrauterine devices (IUDs). ECPs can be taken up to 3–5 days after unprotected sex, but the earlier they are taken, the more effective they are. Emergency contraception pills are Levonorgestrel^©^ (LNG) and ulipristal acetate (UPA). Ulipristal acetate is a selective progesterone receptor modulator. It is a single pill containing 30 mg of UPA, and it is indicated for use up to 120 h after unprotected sexual intercourse [5].

In this case report, we present a case of tubal ectopic pregnancy diagnosed at the eleventh week of gestation in a woman who took ulipristal acetate after unprotected intercourse.

## 2. Clinical Case

A healthy 41-year-old woman, primipara, reported unprotected sexual intercourse and subsequent taking of ulipristal acetate (UPA) the day after. She denied significant medical and obstetrics history, and she declared to have regular menses. She had no risk factors for ectopic pregnancy and only had a laparoscopic cholecystectomy about twenty years earlier. After approximately 11 weeks, she complained of abdominal pain, diarrhea, and syncope, which were treated as gastroenteritis by emergency medical services. Due to menstrual delay, she went to the local Family Contraceptive Clinic to eventually date the pregnancy and plan the voluntary termination of pregnancy. During the examination, the transvaginal ultrasonography revealed the absence of an intrauterine pregnancy and the presence of ectopic pregnancy. For this reason, the patient was then referred to our department.

When she came to our institution, her vital signs were stable. She showed no sign or symptoms of rupture of the ectopic pregnancy. Blood samples were in range, and there were no signs of shock. The physical examination revealed a soft abdomen with no tenderness or rigidity. The ultrasound examination demonstrated a gestational sac in the right tubal area, containing an embryo of 10 + 4 weeks’ (Crown Rump Length of 36.7 mm) (Figure 1). Fetal heartbeat was present. No intrauterine pregnancy was noted. There was a 35 mm layer of free fluid in the pouch of Douglas. Since the diagnosis was already evident, the beta-HCG levels were not measured.

Due to the clinical course, the gestational age, and the size of the ectopic pregnancy, it was decided to take a surgical approach. The laparoscopic evaluation showed 700 mL of hemoperitoneum and an intact but swollen and bleeding right tuba, on the verge of breaking at the level of the ampulla (Figure 2). Right salpingectomy was performed, and the specimen removed was sent for histological examination. The exam confirmed the diagnosis of ectopic pregnancy with embryonic residues and chorionic villi compatible with the first trimester of pregnancy.

The patient recovered without complications, did not need blood transfusions, and was discharged home one day postoperatively.

## 3. Discussion

Ectopic pregnancy is a frequent condition (2% of the pregnancies), and the most common site of implantation is the fallopian tube [2]. Other involvements, such as abdominal, cervical, ovarian, and caesarian section scar, are reported less frequently.

We reported a case of a large ectopic pregnancy in the tubal ampulla at 10 weeks and 4 days of gestation with the presence of fetal heart beating. In the literature, other cases of advanced ectopic pregnancy have been published in other implantation sites. Abdominal implantation sites present increased free space and more ability to distend, allowing larger unruptured ectopic pregnancies. Cases of advanced tubal pregnancy are less frequently described. A report published in 2008 described a ruptured tubal ectopic pregnancy of 11 weeks 1 day of gestation with a beta-HCG over 155.000 IU/L [6]. Other papers describe advanced ectopic pregnancies in the fallopian tube, but no one has found an embryo at over 10 weeks of gestation [7,8].

A factor that makes our case report remarkable is the presence of ectopic pregnancy after ulipristal acetate administration due to unprotected sexual intercourse. Ulipristal acetate is a selective progesterone receptor modulator. In many studies, this molecule has been found to be the most effective oral emergency contraception method, with a failure rate of 1.36%, which does not increase over time if taken in the first 120 h from unprotected intercourse. Furthermore, UPA has been shown to shift the ovulatory phenomenon even to already mature follicular dimensions and to still be able to act even close to the ovulation, maintaining the ability to interfere with ovulation even when the LNG is no longer able to do it [9].

In the literature, few case reports of ectopic pregnancy following emergency contraception are published [10,11]. In these case reports, patients presented ectopic pregnancy after the failure of emergency contraception with Levonorgestrel. As far as we know, no case report of ectopic pregnancy after ulipristal acetate administration is present in the literature. Nevertheless, a 2010 review demonstrated that the rate of ectopic pregnancy when treatment with emergency contraceptive pills fails does not exceed the rate observed in the general population [12]. However, ectopic gestation in case of failure is a known risk.

## 4. Conclusions

This case report shows a rare case of advanced tubal ectopic pregnancy after the failure of emergency contraception with ulipristal acetate. In our opinion, it is important to perform a pregnancy test 20 days after emergency contraception or if menstruation does not reappear. Since emergency contraception is known to fail in some cases, ectopic pregnancy is a concrete risk. In our country, emergency contraception is not only prescribed by gynecologists. For these reasons, women should be informed about the risks and guided in the diagnosis process in case of failure.

## Figures and Tables

**Figure 1 healthcare-10-01590-f001:**
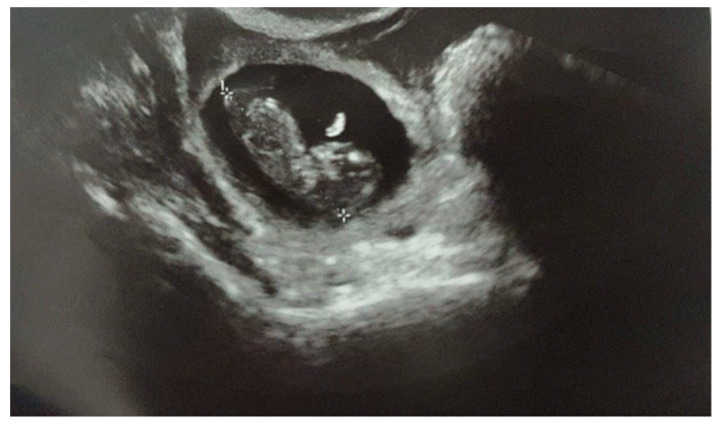
Ultrasound examination demonstrating a 10 + 4 weeks’ tubal pregnancy with a fetal heartbeat.

**Figure 2 healthcare-10-01590-f002:**
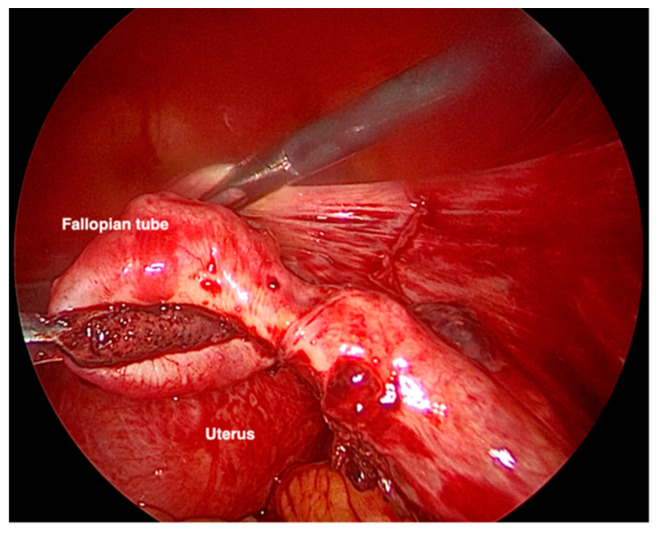
VLS image of the salpinx enlarged and bleeding due to ectopic pregnancy.

## Data Availability

The data that support the findings of this study are available from the corresponding author, upon request.

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
