# Peer review of "A Case of Advanced Tubal Ectopic Pregnancy after Emergency Contraception"

_healthcare, 2022, doi:10.3390/healthcare10081590_

Round 1

Reviewer 1 Report

The authors present a case report of an advanced tubal ectopic pregnancy after emergency contraception with ulipristal acetate. Due to the estimated gestation age of 10 weeks and the clinical presentation, they treated this with a right salpingectomy. 

The authors present an interesting case report which should be of interest to the readership. I would suggest copy editing by a native English speaker to improve the grammar/word choice/punctuation.

More scientific word choices could be used. For example, I would suggest the use of the word "specimen" rather than "surgical piece," and "menstruation" rather than "period." 

Reviewer 2 Report

Dear Editor, dear authors, 

Thank you for giving me the opportunity to review the manuscript entitled "A Case of Advanced Tubal Ectopic Pregnancy after Emergency Contraception". It is a rare case report given the lack of studies on the prevalence of ectopic pregnancies after emergent contraception with UPA. It is also of clinical importance and it should be highlighted that clinicians should be aware of the probability of a failed emergent contraception and the respective risks for the patient. The manuscript has however some shortcomings that preclude publication in its current form. 

Some of them are listed below

Introduction

Lines 42-48: The whole paragraph lack of appropriate references.

Clinical case

Line 81: What about the b-HCG levels?

Lines 89-90: So, did the authors found a ruptured tubal pregnancy? What about the preoperative clinical condition of the patient? The preoperative blood values including haemoglobin and hematocrit as well as vital signs could be useful. 

Reviewer 3 Report

Dear authors, 

This study describes a case of advanced tubal ectopic pregnancy following emergency contraception.

A few general questions and suggestions for the paper:

1. I would request the authors to discuss the incidence of failure of emergency contraception with ullipristal acetate, the mechaisms for failure and the relevency to this case. Additionally, why the authors think these mechanisms led to an ectopic pregnancy and not an intrauterine pregnancy in this patient.

2. Very little  information is given regarding the patient. Was this a spontaneous pregnancy? Does the patient have regular menses?

3. Did the patient have risk factors for ectopic pregnancy? Previous pelvic infections, surgeries, smoking or other.

4. What advice was given to this (and other patients with a similar presentation) regarding future pregnancies?

The following points relate to lines in the manuscript:

11 - The sentence "most cases are diagnosed in initial pregnancy" is unclear - are the authors insinuating that primiparous women have ectopic pregnancies?

32 - "suspect" - change to "suspicious"

34 - "dosage" - change to "level"

34 - there are several possible sonographic presentations for tubal ectopic pregnancy, not just non-cystic adnexal mass, and several sonographic findings with different sensitivity and specificity for diagnosis - I would request the authors elaborate on this.

36 - not HCG - B-hCG

49 - this sentence needs rephrasing, it is unclear.

54 - I request the authors to elaborate on this. When the contralateral tube is abnormal, how would this affect the decision to perform salpingectomy?

58 - "few days" - change to "the days following"

75 - a very brief desciption of the original ultrasound is provided, I would request from the authors to expand upon the initial sonographic findings, what was the BHCG level? This paper is based on a case report, and as such all details of the case must be provided.

Minor grammatical errors should be addressed throughout the paper.

Good luck and thank you for sharing your experience.

Round 2

Reviewer 3 Report

Dear Authors,

Thank you for addressing the issues in my previous review.

In this present form I find the paper adequate.

I would suggest in future reviews that in your cover letter you include reference to page and line numbers for each point in addition to copying the edited text in order to simplify the review process for reviewers.

Good luck.